# Subcutaneous sarilumab for the treatment of hospitalized patients with moderate to severe COVID19 disease: A pragmatic, embedded randomized clinical trial

Westyn Branch-Elliman[1,2]*, Ryan Ferguson[1,3], Gheorghe Doros[1,4], Patricia Woods[1], Sarah Leatherman[1], Judith Strymish[1], Rupak Datta[5,6], Rekha Goswami[7], Matthew D. Jankowich[8], Nishant R. Shah[9], Thomas H. Taylor[10], Sarah T. Page[1], Sara J. Schiller[1], Colleen Shannon[1], Cynthia Hau[1], Maura Flynn[1], Erika Holmberg[1], Karen Visnaw[1], Rupali Dhond[1,3], Mary Brophy[1], Paul A. Monach[1]

1 VA Boston Healthcare System, Boston, Massachusetts, United States of America, 2 Harvard Medical School, Boston, Massachusetts, United States of America, 3 Evans Department of Medicine, Section of General Internal Medicine, Boston University School of Medicine, Boston, Massachusetts, United States of America, 4 Department of Biostatistics, Boston University School of Public Health, Boston, Massachusetts, United States of America, 5 Hospital Epidemiology and Infection Prevention Program, VA Connecticut Healthcare System, West Haven, Connecticut, United States of America, 6 Section of Infectious Diseases, Yale School of Medicine, New Haven, Connecticut, United States of America, 7 Section of Infectious Diseases, VA Maine Healthcare System, Togus, Maine, United States of America, 8 Division of Pulmonary and Critical Care Medicine, Department of Medicine, Providence VA Medical Center, Alpert Medical School of Brown University, Providence, Rhode Island, United States of America, 9 Division of Cardiology, Department of Medicine, Providence VA Medical Center, Alpert Medical School of Brown University, Providence, Rhode Island, United States of America, 10 Infectious Diseases and Rheumatology, White River Jct. VA Medical Center, Hartford, Vermont, United States of America

* wbranche@bidmc.harvard.edu

## Abstract

### Importance and objective

The aim of this pragmatic, embedded, adaptive trial was to measure the effectiveness of the subcutaneous anti-IL-6R antibody sarilumab, when added to an evolving standard of care (SOC), for clinical management of inpatients with moderate to severe COVID-19 disease.

### Design

Two-arm, randomized, open-label controlled trial comparing SOC alone to SOC plus sarilumab. The trial used a randomized play-the-winner design and was fully embedded within the electronic health record (EHR) system.

### Setting

5 VA Medical Centers.

### Participants

Hospitalized patients with clinical criteria for moderate to severe COVID-19 but not requiring mechanical ventilation, and a diagnostic test positive for SARS-CoV-2.

**Data Availability Statement:** All relevant data are within the paper and its Supporting information files.

**Funding:** WBE was supported by NIH NHLBI 1K12HL138049-01. This material is the result of work supported with resources and the use of facilities the VISN-1 Clinical Trials Network and the VA Boston Healthcare System. The funders had no role in study design, data collection and analysis, decision to publish, or preparation of the manuscript.

**Competing interests:** WBE, PM, and JMS were site investigators for a study funded by Gilead Sciences (funds to institution). WBE was supported by NIH NHLBI 1K12HL138049-01. All other authors report no conflicts of interest to report. This does not alter our adherence to PLOS ONE policies on sharing data and materials.

## Interventions

Sarilumab, 200 or 400 mg subcutaneous injection. SOC was not pre-specified and could vary over time, e.g., to include antiviral or other anti-inflammatory drugs.

## Main outcomes and measures

The primary outcome was intubation or death within 14 days of randomization. All data were extracted remotely from the EHR.

## Results

Among 162 eligible patients, 53 consented, and 50 were evaluated for the primary endpoint of intubation or death. This occurred in 5/20 and 1/30 of participants in the sarilumab and SOC arms respectively, with the majority occurring in the initial 9 participants (3/4 in the sarilumab and 1/5 in the SOC) before the sarilumab dose was increased to 400 mg and before remdesivir and dexamethasone were widely adopted. After interim review, the unblinded Data Monitoring Committee recommended that the study be stopped due to concern for safety: a high probability that rates of intubation or death were higher with addition of sarilumab to SOC (92.6%), and a very low probability (3.4%) that sarilumab would be found to be superior.

## Conclusions and relevance

This randomized trial of patients hospitalized due to respiratory compromise from COVID-19 but not mechanical ventilation found no benefit from subcutaneous sarilumab when added to an evolving SOC. The numbers of patients and events were too low to allow definitive conclusions to be drawn, but this study contributes valuable information about the role of subcutaneous IL-6R inhibition in the treatment of hospitalized COVID-19 patients. Methods developed and piloted during this trial will be useful in conducting future studies more efficiently.

## Trial registration

Clinicaltrials.gov—NCT04359901; https://clinicaltrials.gov/ct2/show/NCT04359901?cond=NCT04359901&draw=2&rank=1.

## Introduction

SARS-CoV-2 infection causes the clinical syndrome COVID-19, in which viral pneumonia progresses to respiratory and multi-system organ failure in a subset of patients. Timing of symptom evolution severe enough to require hospitalization in cases of COVID-19, combined with very high levels of C-reactive protein (CRP) and interleukin-6 (IL-6), suggested that the life-threatening manifestations of the disease may be caused by an uncontrolled inflammatory response rather than from a direct viral effect. Blockade of the IL-6 receptor (IL-6R) is an effective treatment in several other conditions featuring excess release of many cytokines [1–3]. Monoclonal antibodies to IL-6R, such as tocilizumab or sarilumab, were of great interest as potential treatments for severe COVID-19; however, enthusiasm about their use in advance of

strong evidence was tempered by concerns about the increased risk of bacterial infections in patients receiving the drug long-term.

Multiple clinical trials using anti-IL-6Rs have been or are still being conducted. These trials included different anti-IL-6Rs, doses, eligibility criteria, outcome measures, and simultaneous use of other treatments, yielding different results that are not yet possible to reconcile. Thus, the role of anti-IL-6Rs, and how and when to administer them, for patients with COVID-19 remains unclear [4–14]. Despite this uncertainty, many medical centers in the US and other countries incorporated varying degrees of off-label use of many drugs [15], including anti-IL-6Rs, as clinicians sought to provide effective treatment based on data that were anecdotal early in the pandemic and remain uncertain even following the publication of multiple trials over the course of a year.

When it became clear by February, 2020, that COVID-19 would become a pandemic disease, programs developed within the Veterans Affairs (VA) New England VA Healthcare System (Veterans Integrated Service Network-1) and the VA Cooperative Studies Program were aligned to provide access to an open-label, pragmatic, adaptive multicenter randomized clinical trial embedded within the VA electronic health record (EHR) with objective outcomes obtainable from chart review. The ten-day time period from initial concept to first patient randomized, including Institutional Review Board (IRB) review and approval and development and deployment of informatics tools to conduct the trial [16], avoided off-label use of unproven therapies while providing clinicians and patients a treatment option that would generate knowledge regarding use of and anti-IL-6R therapy in our veteran population. Here, we present the results of the trial and lessons learned for design of a prospective rapid learning healthcare system [17] to advance evidence generation and translation of evidence into practice.

## Methods

### Study sites

Patients were enrolled from five Veterans Affairs (VA) Medical Centers within VISN-1 with acute medical inpatient services in the Northeast US during the period from April 10, 2020 to February 3, 2021. The study protocol and subsequent amendments were approved by the Institutional Review Boards (IRBs) at each site. All aspects of study management, including regulatory aspects and data collection and analysis, were conducted by a single coordinating center at the VA Boston Healthcare System (VABHS). The trial completed registration at Clinicaltrials.gov 10 days after the study was opened, so as not to delay enrollment under medically urgent circumstances in the setting of a public health emergency. All ongoing and related trials for this drug/intervention are currently registered.

### Study design

This study is an open-label, adaptive, pragmatic randomized trial embedded within the VA Healthcare System EHR using a process similar to previously described Point of Care Clinical Trials [18]. Per pre-specified plans for a play-the-winner design, the first 30 patients were randomized 1:1 to sarilumab or no additional treatment beyond the current SOC. SOC was determined by the treating physicians and local treatment guidance and not pre-determined by study investigators. The randomization ratio was then adapted after assessment of the primary outcome in the first 30 patients and again after each additional 15-patient block. Following a pre-specified scheme, the study statistician determined the change in randomization ratio based on the observed results (See study protocol, S1 and S2 Files). The randomization list was sent to the VA Information Technology Department and then embedded within the VA

electronic health record system. Neither the allocation ratio nor randomization list was shared with the study team. Randomization was not stratified according to recruitment site. Investigators were blinded to the individual patient outcomes, aggregate outcomes, and the randomization ratio after adaptation.

### Intervention

The initial dose of sarilumab (Kevzara) was the FDA-approved dose (200 mg) delivered subcutaneously using the commercially available pre-filled syringe. After the first 9 patients were enrolled, the manufacturer announced that it was discontinuing the 200 mg arm of its own trial of sarilumab based on early data suggesting lack of efficacy [19], thus the dose in this study was increased to 400 mg, delivered as two simultaneous doses of 200 mg subcutaneously.

### Eligibility criteria

Inclusion criteria included a positive SARS-CoV-2 diagnostic test (either PCR or antigen testing) no more than 4 weeks prior to enrollment, presence of symptoms of <14 days duration prior to enrollment, and hospitalization with moderate COVID-19 disease, defined using the Brescia COVID-19 respiratory severity score (BCRSS, subsequently modified) [20].

Exclusion criteria included critical COVID-19, defined by mechanical ventilation and/or expected death within 24 hours; pregnancy; enrollment in another interventional clinical trial; and chronic administration of certain immunosuppressive drugs (e.g., chronic prednisone > 10 mg/day, JAK inhibitors, or immunosuppressive biologics). Short-term use of glucocorticoids and use of any other drug outside of an interventional trial for COVID-19 were permitted.

Details of the changes in eligibility criteria for the purpose of clarifying original intent are in the first and last versions of the protocols used for enrollment (S1 and S2 Files). The most substantive change was in respiratory parameters, which were relaxed during the conduct of the trial. Original criteria required a minimum of 1 of 4 items from the BCRSS: wheezing or inability to speak complete sentences without effort, respiratory rate >22, O2 saturation <90%, or worsening chest X-ray on repeat testing. After enrollment of the first 9 patients, the oxygenation criteria were relaxed to "O2 saturation ≤94% with or without oxygen supplementation, or requiring ≥2L supplemental oxygen to maintain O2 Sat >94% in patients without previously documented hypoxia or baseline oxygenation requirement all within a 24-hour period prior to enrollment," or "worsening of baseline oxygenation by at least 3%, or increase in oxygen requirement by at least 2L, in patients with pre-existing hypoxemia or receiving supplemental oxygen chronically." These changes facilitated determining oxygenation status from the medical record and were more consistent with the earliest large trials that were being published [5,21].

### Primary and secondary outcomes

The primary outcome was a composite of intubation or death within 14 days following randomization. The key secondary outcome was the composite of intubation or death within 30 days after enrollment. Additional data collected are reported in Supplementary Materials.

### Data collection

Screening of patients for eligibility, informed consent with the patient or legally authorized representative, and collection of information about serious adverse events (SAEs) and deaths through day 30 after randomization were all performed by staff at participating sites. The

coordinating center relied on sites to identify and report SAEs and deaths to their local IRBs. Review of these reports of SAEs and deaths was used to adjudicate cause and relationship to COVID-19 by two reviewers in parallel, with additional chart review if needed. All other data were collected remotely through the EHR.

## Statistical analysis

A Bayesian approach using a Beta-Binomial conjugate model was employed to formally assess the superiority of sarilumab to SOC, and to adaptively change the randomization ratio to favor the treatment arm with better outcomes (For details, see original and amended protocols in the S1 and S2 Files). The original randomization ratio was 1:1. In an effort to limit the numbers of patients who would be enrolled using the old ratio while awaiting analysis and adaptation to a new ratio, the primary endpoint was assessed 7 days after the 30[th] patient was enrolled and every additional 15[th] patient thereafter.

Statistical analyses for the primary endpoint were conducted using a statistical test for superiority of proportions based on posterior probability. Within each treatment group, a Beta (3, 12) prior was assumed for the true primary endpoint rates. This prior was selected to have a mean of 20%, to match the average of the assumed rates under the alternative hypothesis and express *a priori* skepticism of the alternative hypothesis being true. The sarilumab group would be declared superior to the SOC group if the posterior probability of the alternative hypothesis being true were larger than 95%. The sarilumab group would be declared inferior to the SOC group if the probability of the true event rate with sarilumab being larger than the rate with SOC plus 3% were larger than 95%. Assuming true primary event rates of 30% and 10% with the SOC and Active treatment group, respectively, a maximum total of 120 subjects, adaptively assigned to treatment A or C yielded 85.7% power to reject the null hypothesis in favor of the alternative.

Statistical analysis was not conducted on secondary outcomes due to high likelihood of types 1 and 2 error due to multiple comparisons between small groups.

## Ethical considerations

IRBs that reviewed and approved the study at each site were the VA Boston Healthcare System Institutional Review Board; Veteran's Institutional Review Board of Northern New England; Institutional Review Board VA Medical Center, Providence RI; and VA Connecticut Healthcare System Human Studies Subcommittee, respectively. The first and final versions of the informed consent documents are available as supplementary materials. After clarifications to the initial US Food and Drug Administration (FDA) guidance were issued [22], written documentation of informed consent was obtained from all participants or their legally authorized representative. Details of the informed consent processes, which were developed to minimize research staff exposure and also limit the use of personal protective equipment at a time when resources were scarce, are previously published [23].

## Results

Chart review screening of 417 patients identified 162 potentially eligible participants; 53 consented and 50 were randomized and evaluated for the primary endpoint (Fig 1). Data shown for these 50 patients include demographic and clinical data (Table 1), relevant comorbidities (Table 2), relevant concomitant medications (Tables 3 and S1), baseline symptoms that could be assessed from the EHR (S2 Table), and maximally abnormal lab tests assessed between hospital arrival and enrollment (S3 Table).

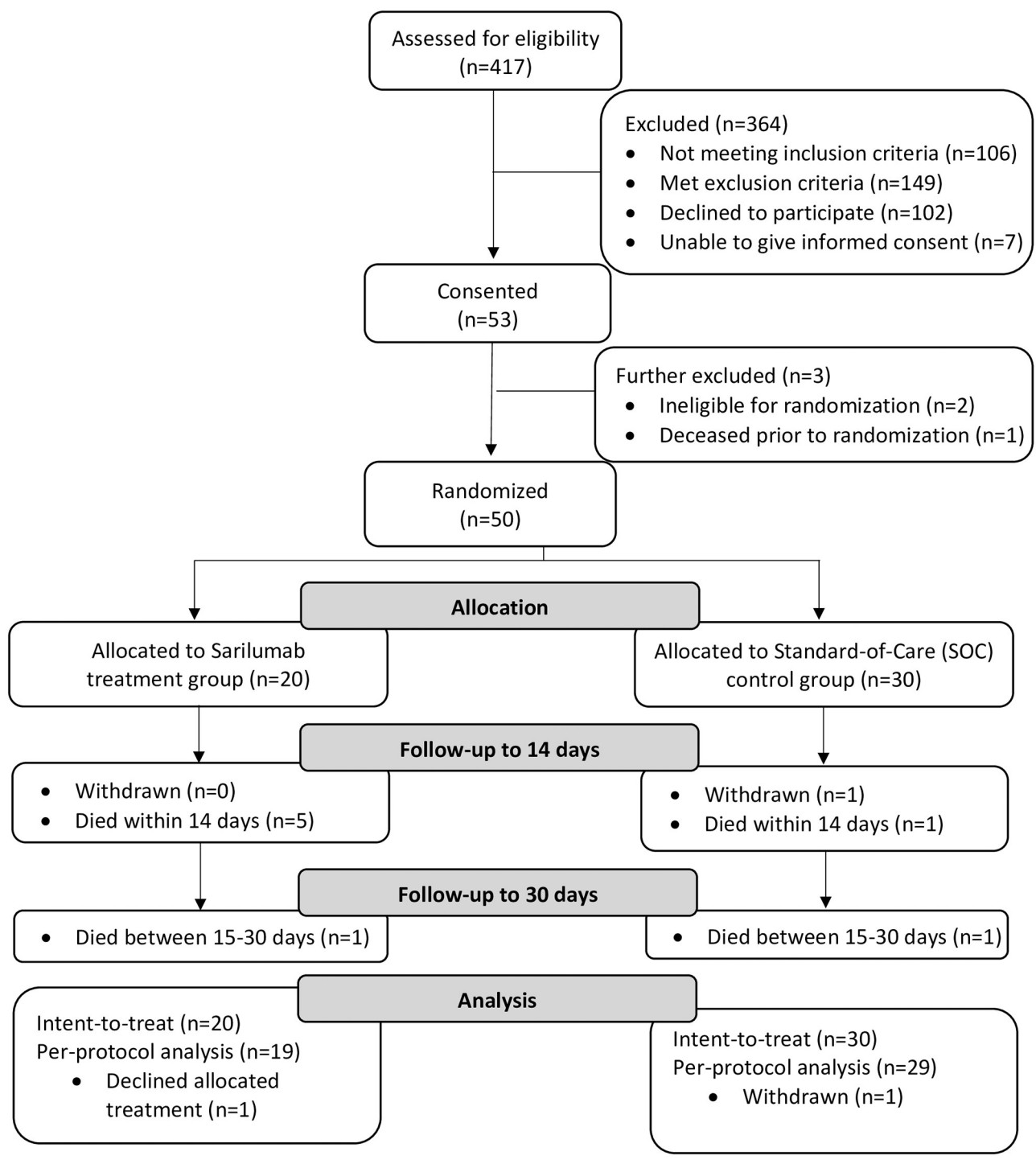

**Fig 1. Consort diagram.**

Nine patients were enrolled between April 10–24, 2020, when the original eligibility criteria required more severe hypoxemia, and neither remdesivir nor dexamethasone was used concomitantly, and the dose of sarilumab was 200 mg. The remaining 41 patients were enrolled between October 8, 2020, and February 3, 2021, after criteria for hypoxemia had been relaxed

**Table 1. Demographics and baseline information.**

| Characteristics[0] | Sarilumab (N = 20) | SOC (N = 30) | Total (N = 50) |
|---|---|---|---|
| Age (years) | 74.8 ± 8.5 | 70.7 ± 14.8 | 72.3 ± 12.7 |
| BMI (kg/m$^2$) | 32.4 ± 6.4 | 32.9 ± 6.9 | 32.7 ± 6.6 |
| Sex, n(%) | | | |
| Female | 2 (10.0%) | 2 (6.7%) | 4 (8.0%) |
| Male | 18 (90.0%) | 28 (93.3%) | 46 (92.0%) |
| Race, n(%) | | | |
| Black or African American | 3 (15.0%) | 1 (3.3%) | 4 (8.0%) |
| Caucasian | 17 (85.0%) | 28 (93.3%) | 45 (90.0%) |
| Not reported | 0 (0%) | 1 (3.3%) | 1 (2.0%) |
| Ethnicity, n(%) | | | |
| Hispanic or Latino | 1 (5.0%) | 1 (3.3%) | 2 (4.0%) |
| Not Hispanic or Latino | 18 (90.0%) | 28 (93.3%) | 46 (92.0%) |
| Not reported | 1 (5.0%) | 1 (3.3%) | 2 (4.0%) |
| Smoking status, n(%) | | | |
| Current | 3 (15.0%) | 3 (10.0%) | 6 (12.0%) |
| Former | 11 (55.0%) | 21 (70.0%) | 32 (64.0%) |
| Never | 6 (30.0%) | 5 (16.7%) | 11 (22.0%) |
| Unknown | 0 (0%) | 1 (3.3%) | 1 (2.0%) |
| Brescia COVID respiratory severity score[1] | 1.8 ± 0.6 | 1.7 ± 0.7 | 1.7 ± 0.7 |
| Body temperature[2] | 98.3 ± 1.3 | 98.3 ± 1.2 | 98.3 ± 1.2 |
| Minimum oxygen saturation on room air[2] | 93.4 ± 2.7 | 93.7 ± 2.9 | 93.5 ± 2.8 |
| Use of Oxygen Supplementation[2], n(%) | 11 (55.0%) | 15 (50.0%) | 26 (52.0%) |
| Oxygen concentrator flow rate[2] (L/min) | 2.7 ± 1.3 | 2.6 ± 0.8 | 2.7 ± 1.1 |
| Non-invasive ventilatory support[2], n(%) | 0 (0%) | 0 (0%) | 0 (0%) |

[0] Reporting mean +/- SD unless otherwise noted.

[1] Evaluated within a 24-hour period prior to randomization;

[2] determined as the closest measurement collected from 24-hour before COVID admission to randomization.

and the sarilumab dose doubled; in addition, 36 and 35 of these patients received remdesivir and dexamethasone, respectively. The randomization scheme was modified twice on the basis of results obtained up to that point in the study, after assessment of the primary endpoint in 30 and 45 patients.

The primary endpoint of intubation or death within 14 days occurred in 5/20 patients in the sarilumab group and 1/30 in the SOC group (Tables 4 and 5); at 30 days, intubation or death had occurred in 6/20 in the sarilumab group and 2/30 in the SOC group. Four deaths occurred among the 9 patients randomized prior to dose modification (3/4 in the sarilumab arm vs. 1/5 in the SOC arm). In the later phase, the 30-day primary outcome occurred in 3/16 patients in the sarilumab group and 1/25 in the SOC group. Of 8 deaths, 4 were attributed directly to progression of respiratory failure from COVID-19 in patients with DNI status, and 2 occurred in patients requiring intubation for respiratory failure within 2 days of enrollment who subsequently developed multi-system organ failure. The 2 other deaths occurred in patients who had been discharged to skilled nursing facilities; one patient had severe exacerbation of dementia and died of dehydration and renal failure under instructions for no aggressive care, and the cause of the other death could not be determined.

Based on these outcomes, the probability of sarilumab being superior to SOC was 3.4%, and the probability of sarilumab being inferior to SOC was 92.6%, i.e. close to the pre-specified

**Table 2. Medical history.**

| Comorbid condition[0] | Sarilumab (N = 20) | SOC (N = 30) | Total (N = 50) |
|---|---|---|---|
| Hypertension | 18 (90.0%) | 25 (83.3%) | 43 (86.0%) |
| Obesity | 15 (75.0%) | 16 (53.3%) | 31 (62.0%) |
| Diabetes mellitus | 12 (60.0%) | 13 (43.3%) | 25 (50.0%) |
| Chronic lung disease | 6 (30.0%) | 15 (50.0%) | 21 (42.0%) |
| Asthma | 3 (15.0%) | 5 (16.7%) | 8 (16.0%) |
| Bronchiectasis | 1 (5.0%) | 2 (6.7%) | 3 (6.0%) |
| Chronic obstructive pulmonary disease | 4 (20.0%) | 12 (40.0%) | 16 (32.0%) |
| Cardiovascular disease | 13 (65.0%) | 22 (73.3%) | 35 (70.0%) |
| Cardiac arrhythmia | 12 (60.0%) | 16 (53.3%) | 28 (56.0%) |
| Coronary artery disease | 11 (55.0%) | 15 (50.0%) | 26 (52.0%) |
| Congestive heart failure | 7 (35.0%) | 8 (26.7%) | 15 (30.0%) |
| Renal disease | 6 (30.0%) | 3 (10.0%) | 9 (18.0%) |
| Liver disease | 2 (10.0%) | 1 (3.3%) | 3 (6.0%) |
| Thrombocytopenia | 2 (10.0%) | 0 (0%) | 1 (2.0%) |
| Rheumatologic and autoimmune disease | 1 (5.0%) | 2 (6.7%) | 3 (6.0%) |
| Rheumatoid arthritis | 1 (5.0%) | 0 (0%) | 1 (2.0%) |
| Autoimmune disease | 0 (0%) | 2 (6.7%) | 2 (4.0%) |
| Neurologic disease | 8 (40.0%) | 5 (16.7%) | 13 (26.0%) |
| Cancer | 10 (50.0%) | 14 (46.7%) | 24 (48.0%) |
| Solid tumor malignancies | 9 (45.0%) | 14 (46.7%) | 23 (46.0%) |
| Hematologic malignancies | 1 (5.0%) | 0 (0%) | 1 (2.0%) |
| Psychiatric disorder | 11 (55.0%) | 13 (43.3%) | 24 (48.0%) |
| Alcohol use disorder | 2 (10.0%) | 5 (16.7%) | 7 (14.0%) |
| Nicotine dependance | 8 (40.0%) | 8 (26.7%) | 14 (28.0%) |

[0] Comorbidities were determined based on ICD-9-CM and ICD-10-CM diagnosis codes assigned before the randomized date.

stopping rule for futility (97.5%) but not crossing it. Excluding data from the first 9 patients, the probability of 400 mg subcutaneously-delivered sarilumab being superior was 11.2%, and the probability of being no more than 3% superior was 78.6% (Tables 5 and S6).

Because of the small number of events and the lack of benefit seen with sarilumab, data on the primary endpoint were not adjusted for other risk factors potentially related to outcome. Data on clinical, laboratory, and medication parameters in patients who did or did not meet

**Table 3. Medication administration during COVID hospitalization by drug categories.**

| Medications | Sarilumab (N = 20) | SOC (N = 30) | Total (N = 50) |
|---|---|---|---|
| ACE | 8 (40.0%) | 8 (26.7%) | 16 (32.0%) |
| Antibiotics | 17 (85.0%) | 29 (96.7%) | 46 (92.0%) |
| Anticoagulants | 20 (100.0%) | 30 (100.0%) | 50 (100.0%) |
| Glucocorticoids | 17 (85.0%) | 26 (86.7%) | 43 (86.0%) |
| Statins | 14 (70.0%) | 19 (63.3%) | 33 (66.0%) |
| Convalescent plasma | 0 (0%) | 1 (3.3%) | 1 (2.0%) |
| Hydroxychloroquine | 1 (5.0%) | 1 (3.3%) | 2 (4.0%) |
| Remdesivir | 15 (75.0%) | 25 (83.3%) | 40 (80.0%) |
| Rituximab | 0 (0%) | 1 (3.3%) | 1 (2.0%) |
| Tocilizumab | 0 (0%) | 1 (3.3%) | 1 (2.0%) |

**Table 4. Study outcome.**

| Primary and secondary outcome[0] | Sarilumab (N = 20) | SOC (N = 30) |
|---|---|---|
| Composite primary outcome occurred within 14 days | 5 (25.0%) | 1 (3.3%) |
| Intubation | 2 (10.0%) | 0 (0%) |
| Death without prior intubation | 3 (15.0%) | 1 (3.3%) |
| Secondary outcome occurred within 30 days | | |
| All-cause mortality | 6 (30.0%) | 2 (6.7%) |
| Oxygen saturation recovery[1] | 10 (50.0%) | 15 (50.0%) |
| Intensive care unit admission | 3 (15.0%) | 3 (10.0%) |
| New onset of delirium | 0 (0%) | 1 (3.3%) |
| New onset of heart failure | 2 (10.0%) | 0 (0%) |
| New onset of arrhythmia | 1 (5.0%) | 1 (3.3%) |
| New or worsening renal failure | 3 (15.0%) | 2 (6.7%) |
| Thromboembolic disease | 1 (5.0%) | 0 (0%) |
| Patient discharged | 15 (80.0%) | 27 (90.0%) |
| Clinical outcome status[2] | | |
| At time of randomization (n = 49) | 4.9 ± 0.6 | 5.0 ± 0.5 |
| At time of discharge (n = 43) | 4.3 ± 0.4 | 4.4 ± 0.6 |
| At 30 days from randomization (n = 41) | | |
| <4 | 13 (65.0%) | 26 (86.7%) |
| ≥4 | 1 (5.0%) | 1 (3.3%) |

[0] Adjudicated outcome captured within 30 days from randomization date.

[1] Recovery was defined as >94% without out supplemental oxygen or return to the patient's baseline level.

[2] Patient status was evaluated using a 7-point ordinal scale ranging from not hospitalized (score = 1) to death (score = 7).

the primary endpoint are shown in S4 Table. Secondary outcomes also demonstrated no difference between the sarilumab and SOC groups (Table 4).

Four serious adverse events not attributable to COVID-19 occurred: one case of delirium that resolved, one case of transient hypoxemia and tachycardia, one case of exacerbation of heart failure, and one case of acute kidney injury with diagnosis of previously unrecognized amyloidosis due to a B-cell lymphoproliferative disease (treated with rituximab after enrollment). Specific adverse events extrapolated from the safety profile of chronic sarilumab use were not substantially higher in the sarilumab group (S5 Table).

Based on the two interim analyses performed for the adaptive randomization and the full two- week treatment experience of all 50 patients randomized to therapy, the unblinded Data Monitoring Committee recommended that the study be stopped out of concern for the high probability that rates of intubation or death were higher in the sarilumab arm than the SOC arm. While the statistical analysis plan had formal stopping criteria, the committee voted to recommend early trial discontinuation in light of the collected data rather than following the *a priori* thresholds for termination. The committee concluded that the potential harm and the very low probability of showing benefit outweighed the necessity for strict adherence to the *a priori* (and perhaps arbitrary) thresholds from the statistical analysis plan.

## Discussion

This randomized trial of patients hospitalized with COVID-19 with respiratory symptoms but not requiring mechanical ventilation showed no evidence of benefit from subcutaneous

**Table 5. Analysis and adaptation based on the primary outcome.**

| Time | Study Milestone | Sarilumab (Events/Subjects) | Standard of Care (Events/Subjects) |
|---|---|---|---|
| | Randomization Ratio = 50% Sarilumab/50% SOC | | |
| | Study Start until N = 30 Enrolled | 5/14 | 1/15 |
| | First Interim Analysis | | |
| | | Probability Sarilumab Superior = 7.9% | Probability Sarilumab Inferior = 86.8% |
| | Updated Randomization Ratio = 21.9% Sarilumab/78.1% SOC | | |
| | Additional Analyzed after First Interim Analysis | 0/5 | 0/9 |
| | Second Interim Analysis | | |
| | | 5/19 Total | 1/24 Total |
| | | Probability Sarilumab Superior = 5.78% | Probability Sarilumab Inferior = 87.6% |
| | Updated Randomization Ratio = 19.4% Sarilumab/ 80.6% SOC | | |
| | Follow-up after Second Interim Analysis | 0/0 | 0/6 |
| | Third Interim Analysis | | |
| | | 5/19 total/final | 1/30 total/final |
| | | Probability Sarilumab Superior = 3.36% | Probability Sarilumab Inferior = 92.6% |
| | Study Stopped | | |
| | Subset Limited to Dose 400 mg (N = 41) | 2/15 | 0/25 |
| | | Probability Sarilumab Superior = 11.2% | Probability Sarilumab Inferior = 78.6% |

sarilumab, either 200 mg or 400 mg. The numbers of patients and events were too low to allow definitive conclusions to be drawn, but this study still contributes valuable information, as data from many controlled trials are being collected worldwide and could determine which patients might benefit from use of IL-6R blockade with either sarilumab or tocilizumab. A major contribution of this trial–which enrolled its first patient within 10 days of being proposed to the key decision-makers–was its integration into the EHR system and demonstration of the real-time use of a learning healthcare system for prospective clinical trials. The methods developed could improve the efficiency of a wide range of future trials, especially trials of FDA-approved medications for off-label use.

This study advances but cannot reconcile a complex and conflicting literature about the role of IL-6R blockade in the management of severe and critical COVID-19 disease [4–14]. Results from the first 9 patients in the trial are no longer highly relevant, and are difficult to interpret in the setting of substantial advances in the treatment of patients hospitalized with COVID-19, particularly the addition of dexamethasone and remdesivir as part of the SOC. Mortality in this group, which included many old and frail nursing home residents, was very high, as was typical in our region early in the pandemic.

The literature in patients most similar to those enrolled in this study on the basis of disease severity argues for benefit of either tocilizumab or sarilumab given intravenously in combination with remdesivir and dexamethasone [6,7]. Our study suggests–although not definitively–that subcutaneous injection may not be considered an adequate substitute. Additionally, our study adds support to recommendations that if anti-IL-6R therapy is used, relatively high doses (e.g., 400 mg sarilumab or 8 mg/kg tocilizumab) may be required to achieve a therapeutic response. These conclusions remain relevant because COVID-19 remains active around the world, and additional variants are likely to continue to cause waves of infections. The commercially available form of sarilumab is the pre-filled syringe used in our trial, so if the drug is used clinically, IV solutions should be prepared as they were in other trials [6].

When delivered subcutaneously, the peak drug level is achieved approximately 2 days post injection, and thus medication may not be available in a sufficiently short timeframe to alter

disease progression among patients with already severe disease and a severe systemic inflammatory response. High rates of obesity (75%) in our population may also have contributed to limited absorption of the study drug when administered subcutaneously.

Our study does not prove that subcutaneous sarilumab is ineffective or harmful using the conventional criterion of a P-value, but based on posterior probabilities, it provides strong support for the notion that if it works at all, any clinical effect in hospitalized patients without organ failure is limited. Although this is not the philosophical approach typically taken in phase 3 trials, pharmaceutical companies make rational "go versus no-go" decisions early in drug development, and practicing physicians increase or decrease use of drugs–whether approved or off-label–based on their own experience and that of their colleagues.

The major strengths of this study are its design and rapid implementation and deployment, aimed to enroll patients and test its hypothesis as quickly as possible in a real-world clinical setting. A Bayesian design was used to re-calculate probabilities in real-time during the trial. Eligibility criteria and outcomes could be assessed remotely through the EHR, and the coordinating center leveraged experience and tools developed to conduct previous pragmatic multi-center clinical trials.

A strength conferred by adaptive randomization is that, over time, patients are increasingly likely to receive the more beneficial treatment [16,24,25]. Adaptive randomization after a small number of events led to more patients being randomized to SOC. Although others might point to this fact as an example of the deficiency of adaptive randomization, we argue that the point of many clinical trials should be to get to a clinically-actionable answer as rapidly as possible, and to reduce the significant lag between evidence generation and adoption into clinical care, rather than to test a hypothesis in the most definitive and quantifiable way. Currently, there is typically a 17-year lag between evidence generation and translation of evidence into clinical practice [26]. The embedded approach used in this trial is a mechanism that may be leveraged in the future to encourage immediate transitions of evidence generation into implementation, which has been a major challenge in clinical medicine. If sarilumab had been found to be effective, then mechanisms and order sets developed and embedded as part of the research could have been immediately converted into clinical decision support tools, with ongoing collection of data on efficacy and toxicity. The pragmatic and adaptive features developed and implemented in this trial should be applied much more widely, beyond emergency circumstances.

The plan to have the trial "adapt" to changing circumstances and a changing SOC in the setting of a novel and life-threatening disease is both a strength (ethically) and a weakness (scientifically and operationally). In 2 of 10 total amendments, a significant inclusion criterion (degree of hypoxemia) and the dose of study drug were changed, at the same time that underlying SOC changed, so the first 9 patients and the last 41 patients were treated differently. Other trials have changed outcome measures and limited exclusion criteria based on concomitant treatment, for similar and equally appropriate reasons [5–7,21,27,28]. Although the results limited to the period when the 400 mg dose was used, which was also a time when dexamethasone and remdesivir also became widely used, were not entirely convincing for lack of benefit on their own, the probability of benefit was only 11% based on an event rate of 2/15 in the sarilumab arm versus 0/25 in the control arm, which in the context of earlier results was sufficient to stop the trial.

The workload generated by parallel review of the study, including the original documents and 10 amendments, at 5 separate sites is a vindication of the stipulation in the revised Common Rule that multi-site studies should use a single IRB of record. Factors that facilitated the rapid operationalization of the study included that leaders of the VABHS IRB and Research and Development committees assisted in preparing or reviewing documents on very short

notice; the pharmacy secured study drug during a time of nationwide depletion of anti-IL6R antibodies; and existing infrastructure available through the VISN-1 Clinical Trials Network facilitated study operations at multiple sites. Use of an FDA-approved drug that did not require an IND was also a key factor in opening the trial quickly; there was a substantial pause after the dose increase as a waiver of IND was required.

Although the trial opened quickly, and the remote monitoring and embedding of group assignment and outcomes in the EHR proceeded well, the one time-consuming barrier to efficient conduct of the trial was the difficulty in obtaining signed documentation from severely ill patients of informed consent, as we described in detail elsewhere [23]. The pace of finding effective treatments, especially in emergency circumstances, would be greatly improved by a change in perspective of administrative and regulatory bodies. Rapid response would be permitted by establishment and coordination of clinical trials networks in many countries that could be mobilized on short notice;[27] by use of pragmatic and adaptive designs [6,9,21,27]; by planning ahead for development of protocols and data management and analysis; and by streamlining of processes for informed consent under conditions of quarantine.

## Conclusions

In this multi-center, adaptive, point-of-care randomized controlled trial evaluating the effectiveness of subcutaneous sarilumab added to a changing standard of care, we found no evidence of benefit and possible suggestion of harm. Methods developed represent a realization of the learning healthcare system model and may be applied in other studies of FDA-approved medications for off-label use to advance evidence generation and speed the adoption of new evidence into clinical care.

## Supporting information

**S1 Checklist. CONSORT 2010 checklist of information to include when reporting a randomised trial**\*.
(DOC)

**S1 Table. Full list of medication administration during COVID hospitalization.**
(DOCX)

**S2 Table. Symptomatology on admission.**
(DOCX)

**S3 Table. Laboratory tests ordered between ED arrival and randomization.**
(DOCX)

**S4 Table. Comparison of selected clinical variables between patients who died or survived within 30 days of study period.**
(DOCX)

**S5 Table. Expected adverse events.**
(DOCX)

**S6 Table. Interim analysis details.**
(DOCX)

**S1 File.**
(PDF)

**S2 File.**
(PDF)

## Acknowledgments

This study would not have been possible without the support of the entire inpatient medical staff at the participating sites for their tireless efforts helping the study team identify and screen patients. We would also like to thank the VISN-1 Clinical Trials Network and Dr. William Boden for his support of the trial, the VA Boston Research Pharmacy and Drs. Antoun Houranieh and Jane Hughes for all of their efforts procuring and distributing study medication. We would also like to acknowledge the help and support of Drs. Michael Charness, Lisa Soleymani Lehmann, Carole Palumbo, and David Thornton, as well as the research efforts of Rebecca Anderson, David Ardito, Karen Evans, Jodi Okrant, and Patricia Spencer.

The views expressed in this manuscript are those of the authors and they do not necessarily reflect the views of the United States Federal Government or the Department of Veterans Affairs.

## Author Contributions

**Conceptualization:** Westyn Branch-Elliman, Ryan Ferguson, Gheorghe Doros, Sarah Leatherman, Mary Brophy, Paul A. Monach.

**Data curation:** Judith Strymish, Sarah T. Page, Sara J. Schiller, Cynthia Hau, Erika Holmberg, Rupali Dhond.

**Formal analysis:** Westyn Branch-Elliman, Ryan Ferguson, Gheorghe Doros, Sarah Leatherman, Cynthia Hau, Erika Holmberg, Rupali Dhond, Mary Brophy, Paul A. Monach.

**Investigation:** Westyn Branch-Elliman, Ryan Ferguson, Patricia Woods, Sarah Leatherman, Judith Strymish, Rupak Datta, Rekha Goswami, Matthew D. Jankowich, Nishant R. Shah, Thomas H. Taylor, Sara J. Schiller, Maura Flynn, Erika Holmberg, Karen Visnaw, Rupali Dhond, Mary Brophy, Paul A. Monach.

**Methodology:** Westyn Branch-Elliman, Ryan Ferguson, Gheorghe Doros, Patricia Woods, Sarah Leatherman, Thomas H. Taylor, Sarah T. Page, Sara J. Schiller, Erika Holmberg, Mary Brophy, Paul A. Monach.

**Project administration:** Westyn Branch-Elliman, Ryan Ferguson, Patricia Woods, Sarah Leatherman, Rupak Datta, Matthew D. Jankowich, Sarah T. Page, Sara J. Schiller, Colleen Shannon, Cynthia Hau, Maura Flynn, Erika Holmberg, Karen Visnaw, Rupali Dhond, Mary Brophy.

**Resources:** Westyn Branch-Elliman, Ryan Ferguson, Sarah Leatherman, Judith Strymish, Rupak Datta, Matthew D. Jankowich, Nishant R. Shah, Sarah T. Page, Sara J. Schiller, Colleen Shannon, Cynthia Hau, Karen Visnaw, Rupali Dhond, Paul A. Monach.

**Supervision:** Westyn Branch-Elliman, Erika Holmberg, Mary Brophy, Paul A. Monach.

**Validation:** Sarah T. Page.

**Writing – original draft:** Westyn Branch-Elliman, Ryan Ferguson, Gheorghe Doros, Sarah Leatherman, Sara J. Schiller, Cynthia Hau, Maura Flynn, Erika Holmberg, Karen Visnaw, Mary Brophy, Paul A. Monach.

**Writing – review & editing:** Westyn Branch-Elliman, Ryan Ferguson, Gheorghe Doros, Patricia Woods, Sarah Leatherman, Judith Strymish, Rupak Datta, Rekha Goswami, Matthew D.

Jankowich, Nishant R. Shah, Thomas H. Taylor, Sarah T. Page, Sara J. Schiller, Colleen Shannon, Cynthia Hau, Maura Flynn, Erika Holmberg, Karen Visnaw, Rupali Dhond, Mary Brophy, Paul A. Monach.

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
