## [Decision Letter · Decision Letter 0]

15 Nov 2021

PONE-D-21-20348Subcutaneous Sarilumab for the Treatment of Hospitalized patients with Moderate to Severe COVID19 Disease: A Pragmatic, Embedded Randomized Clinical Trial

PLOS ONE

Dear Dr. Branch-Elliman,

Thank you for submitting your manuscript to PLOS ONE. After careful consideration, we feel that it has merit but does not fully meet PLOS ONE’s publication criteria as it currently stands. Therefore, we invite you to submit a revised version of the manuscript that addresses the points raised during the review process.

We look forward to receiving your revised manuscript.

Kind regards,

Giuseppe Vittorio De Socio, MD, PhD

Academic Editor

PLOS ONE

Journal Requirements:

2. Please provide additional details regarding participant consent. In the ethics statement in the Methods and online submission information, please ensure that you have specified what type you obtained (for instance, written or verbal, and if verbal, how it was documented and witnessed).

3. Thank you for submitting your clinical trial to PLOS ONE and for providing the name of the registry and the registration number. The information in the registry entry suggests that your trial was registered after patient recruitment began. PLOS ONE strongly encourages authors to register all trials before recruiting the first participant in a study.

1) your reasons for your delay in registering this study (after enrolment of participants started);

2) confirmation that all related trials are registered by stating: “The authors confirm that all ongoing and related trials for this drug/intervention are registered”.

 [WBE was supported by NIH NHLBI 1K12HL138049-01. 

This material is the result of work supported with resources and the use of facilities the VISN-1 Clinical Trials Network and the VA Boston Healthcare System.]

[WBE, PM, and JMS were site investigators for a study funded by Gilead Sciences (funds to institution). All other authors have no conflicts of interest to report.]

Reviewers' comments:

Reviewer's Responses to Questions

**Comments to the Author**

1. Is the manuscript technically sound, and do the data support the conclusions?

Reviewer #1: Yes

Reviewer #2: Yes

2. Has the statistical analysis been performed appropriately and rigorously? 

Reviewer #1: Yes

Reviewer #2: Yes

3. Have the authors made all data underlying the findings in their manuscript fully available?

Reviewer #1: Yes

Reviewer #2: No

4. Is the manuscript presented in an intelligible fashion and written in standard English?

Reviewer #1: Yes

Reviewer #2: Yes

5. Review Comments to the Author

Reviewer #1: This is an interesting study. Given the pandemic, the play-the-winner design was appropriate. There are a couple of concerns. One concern is the small sample size based on which the conclusion was reached. Second concern is the initial 200mg dose, and addition of remdesivir and dexamethasone to SOC. These made it difficult to interpret the findings.

Some clarifications are needed as in the specific comments.

Page 6, in section Study design:

“Per pre-specified plans for a play-the-winner design, the first 30 patients were randomized 1:1 to sarilumab or no additional treatment beyond the current SOC, as determined by treating physicians.”

Was the randomization determined by treating physicians or by a randomization scheme?

“Following a pre-specified scheme, the study statistician determined the change in randomization

based on the observed results.”

Please add details on the randomization scheme so that it is clear to the readers.

“Neither the allocation ratio nor randomization list was shared with the study team. Randomization was not stratified according to recruitment site. Investigators were blinded to the results and randomization ratio.”

What does it mean “blinded to the results”? A statistician would infer it mean the investigators were blinded on what treatment the patient was receiving. Please clarify. Did you use placebo to ensure the blinding? If no placebo was used, what’s the reason for that decision? What measure was taken to ensure no bias was introduced when the study endpoint was assessed?

Reviewer #2: This is a well written manuscript. I only have one concern.

The number of primary outcomes listed in Table 4 does not appear to match those listed in the abstract and main text of the Results section, for example, 5 (Table 4) versus 6 (text) in the Sarilumab group.

6. PLOS authors have the option to publish the peer review history of their article (what does this mean?). If published, this will include your full peer review and any attached files.

Reviewer #1: No

Reviewer #2: No

---

## [Author Response · Author response to Decision Letter 0]

18 Nov 2021

Reviewers' comments:

Reviewer's Responses to Questions

 Comments to the Author

1. Is the manuscript technically sound, and do the data support the conclusions?

Reviewer #1: Yes

Reviewer #2: Yes

 2. Has the statistical analysis been performed appropriately and rigorously? 

 Reviewer #1: Yes

Reviewer #2: Yes

 3. Have the authors made all data underlying the findings in their manuscript fully available?

 Reviewer #1: Yes

Reviewer #2: No

Response: Deidentified data will be made available to investigators through a data use agreement, per national VA policy and per what was included in the informed consent. This is clarified in the data availability statement and outlined in our cover letter.

 4. Is the manuscript presented in an intelligible fashion and written in standard English?

 Reviewer #1: Yes

Reviewer #2: Yes

 5. Review Comments to the Author

Reviewer #1: This is an interesting study. Given the pandemic, the play-the-winner design was appropriate. There are a couple of concerns. One concern is the small sample size based on which the conclusion was reached. Second concern is the initial 200mg dose, and addition of remdesivir and dexamethasone to SOC. These made it difficult to interpret the findings.

Response: We appreciate the reviewer’s interest in the study and the justification for its design. We had attempted to fully acknowledge this scientific weakness in the original discussion: 

“The plan to have the trial “adapt” to changing circumstances and a changing SOC in the setting of a novel and life-threatening disease is both a strength (ethically) and a weakness (scientifically and operationally). In 2 of 10 total amendments, a significant inclusion criterion (degree of hypoxemia) and the dose of study drug were changed, at the same time that underlying SOC changed, so the first 9 patients and the last 41 patients were treated differently. Other trials have changed outcome measures and limited exclusion criteria based on concomitant treatment, for similar and equally appropriate reasons.” 

To address the reviewer comment, we added the following language: “Although the results limited to the period when the 400 mg dose was used, which was also a time when dexamethasone and remdesivir also became widely used, were not entirely convincing for lack of benefit on their own, the probability of benefit was only 11% based on an event rate of 2/15 in the sarilumab arm versus 0/25 in the control arm, which in the context of earlier results was sufficient to stop the trial.”

Some clarifications are needed as in the specific comments.

Page 6, in section Study design:

“Per pre-specified plans for a play-the-winner design, the first 30 patients were randomized 1:1 to sarilumab or no additional treatment beyond the current SOC, as determined by treating physicians.” Was the randomization determined by treating physicians or by a randomization scheme?

Response: We agree that the language was confusing. The standard of care was determined by the treating physicians, not the randomization. The section now reads, “Per pre-specified plans for a play-the-winner design, the first 30 patients were randomized 1:1 to sarilumab or no additional treatment beyond the current SOC. SOC was determined by the treating physicians and local treatment guidance and not pre-determined by investigators.”

“Following a pre-specified scheme, the study statistician determined the change in randomizationbased on the observed results.” Please add details on the randomization scheme so that it is clear to the readers.

Response: We agree that more detail is needed for clarity. To address this issue, we now refer to the study protocol, which are included as supplemental documents. Details of the adaptive randomization process are outlined in these documents.

“Neither the allocation ratio nor randomization list was shared with the study team. Randomization was not stratified according to recruitment site. Investigators were blinded to the results and randomization ratio.” What does it mean “blinded to the results”? A statistician would infer it mean the investigators were blinded on what treatment the patient was receiving. Please clarify. Did you use placebo to ensure the blinding? If no placebo was used, what’s the reason for that decision? What measure was taken to ensure no bias was introduced when the study endpoint was assessed?

Response: We apologize for the confusion caused by the way the statements were phrased. Neither the investigators not the patients were blinded to the intervention, but investigators were blinded to aggregate outcomes that determined the adaptive randomization, and also to the updated randomization ratio. Based on reviewer feedback, we have revised the text on page 7 to make the process more clear. We did not take any specific measure to ensure no bias was introduced into the assessment of outcomes; however, we chose only objective outcomes (mechanical ventilation or death) in order to limit the impact of this potential source of bias. Use of placebo would have dramatically increase the cost and regulatory complexity of the study.

Reviewer #2: This is a well written manuscript. I only have one concern.

The number of primary outcomes listed in Table 4 does not appear to match those listed in the abstract and main text of the Results section, for example, 5 (Table 4) versus 6 (text) in the Sarilumab group.

Response: We apologize for reporting the primary outcome differently in different places and have corrected the abstract so that the numbers refer to the outcome occurring within 14 days. The text on p. 11 has been corrected to provide numbers for the 14-day and 30-day outcomes, and deaths were adjudicated based on 30 days. The tables included both 14- and 30-day outcomes and are not updated.

---

## [Decision Letter · Decision Letter 1]

24 Jan 2022

Subcutaneous Sarilumab for the Treatment of Hospitalized patients with Moderate to Severe COVID19 Disease: A Pragmatic, Embedded Randomized Clinical Trial

PONE-D-21-20348R1

Dear Dr. Branch-Elliman,

We’re pleased to inform you that your manuscript has been judged scientifically suitable for publication and will be formally accepted for publication once it meets all outstanding technical requirements.

Kind regards,

Giuseppe Vittorio De Socio, MD, PhD

Academic Editor

PLOS ONE

Additional Editor Comments (optional):

Reviewers' comments:

Reviewer's Responses to Questions

**Comments to the Author**

1. If the authors have adequately addressed your comments raised in a previous round of review and you feel that this manuscript is now acceptable for publication, you may indicate that here to bypass the “Comments to the Author” section, enter your conflict of interest statement in the “Confidential to Editor” section, and submit your "Accept" recommendation.

Reviewer #1: (No Response)

Reviewer #2: All comments have been addressed

2. Is the manuscript technically sound, and do the data support the conclusions?

Reviewer #1: Yes

Reviewer #2: Yes

3. Has the statistical analysis been performed appropriately and rigorously? 

Reviewer #1: Yes

Reviewer #2: Yes

4. Have the authors made all data underlying the findings in their manuscript fully available?

Reviewer #1: Yes

Reviewer #2: Yes

5. Is the manuscript presented in an intelligible fashion and written in standard English?

Reviewer #1: Yes

Reviewer #2: Yes

6. Review Comments to the Author

Reviewer #1: Minor edits:

On page 6: remove word “and” in “…knowledge regarding use of and anti-IL-6R therapy in our veteran population”.

Reviewer #2: (No Response)

7. PLOS authors have the option to publish the peer review history of their article (what does this mean?). If published, this will include your full peer review and any attached files.

Reviewer #1: No

Reviewer #2: No

---

## [Editor Report · Acceptance letter]

8 Feb 2022

PONE-D-21-20348R1 

Subcutaneous Sarilumab for the Treatment of Hospitalized patients with Moderate to Severe COVID19 Disease: A Pragmatic, Embedded Randomized Clinical Trial 

Dear Dr. Branch-Elliman:

I'm pleased to inform you that your manuscript has been deemed suitable for publication in PLOS ONE. Congratulations! Your manuscript is now with our production department. 

Kind regards, 

on behalf of

Dr. Giuseppe Vittorio De Socio 

Academic Editor

PLOS ONE